# Retrieval with Learned Similarities

## Abstract

Retrieval plays a fundamental role in recommendation systems, search, and natural language processing (NLP) by efficiently finding relevant items from a large corpus given a query. Dot products have been widely used as the similarity function in such retrieval tasks, thanks to Maximum Inner Product Search (MIPS) that enabled efficient retrieval based on dot products. However, state-of-the-art retrieval algorithms have migrated to learned similarities. Such algorithms vary in form; the queries can be represented with multiple embeddings, complex neural networks can be deployed, the item ids can be decoded directly from queries using beam search, and multiple approaches can be combined in hybrid solutions. Unfortunately, we lack efficient solutions for retrieval in these state-of-the-art setups. Our work investigates techniques for efficient retrieval with expressive learned similarity functions. We first prove that Mixture-of-Logits (MoL) is a universal approximator, and can express all learned similarity functions. We then demonstrate how to apply MoL to common retrieval tasks in recommendation systems and NLP. We next propose techniques to retrieve the approximate top $K$ results using MoL with a tight bound. We finally compare our techniques with existing approaches, showing that MoL, with a new mutual information-based load balancing loss we propose, sets new state-of-the-art results across heterogeneous scenarios, including sequential retrieval models in recommendation systems and finetuning language models for question answering; and our approximate top-k retrieval with learned similarities outperforms baselines by up to 105× in latency, while achieving > .99 recall rate of exact algorithms.

## CCS Concepts

• **Information systems** → **Similarity measures**; **Top-k retrieval in databases**; **Learning to rank**; **Probabilistic retrieval models**; *Question answering*; *Recommender systems*; *Personalization*; • **Computing methodologies** → *Natural language processing*.

## Keywords

Nearest Neighbor Search, Learned Similarities, Top-K Retrieval, Vector Databases, Recommendation Systems, Question Answering

**ACM Reference Format:**
Anonymous Author(s). 2018. Retrieval with Learned Similarities. In *Proceedings of Make sure to enter the correct conference title from your rights confirmation emai (Conference acronym 'XX)*. ACM, New York, NY, USA, 12 pages. https://doi.org/XXXXXXX.XXXXXXX

## 1 Introduction

Retrieval requires efficient storing, indexing, and querying relevant candidate items represented by high-dimensional vectors. Retrieval is widely used as the initial preprocessing stage for internet applications such as recommendations, search, question answering, and natural language processing that operate over corpus with up to billions of items [5, 10, 16, 28, 33, 35]. In many concrete use cases, such as vector databases [26], the query- and the item- embeddings are learned with deep neural networks in a dual-encoder setup, and dot products are applied on top of such embeddings as the similarity function for measuring relevance.

Despite the popularity of dot products and numerous work done to improve their efficiency [9, 25, 37, 51], state-of-the-art retrieval algorithms have long moved to various learned similarity functions. Their most basic versions preserve some dot product-related structures, but turn either the query or the item into multiple embeddings, and rely on a max operator to combine those similarity values [29, 35]. As another example, Probabilistic Label Trees (PLTs) [23] and Tree-based Deep Models (TDMs) [62, 64] map items to leaf nodes in a tree, and reduce retrieval to beam search by making decisions sequentially using learned classifiers while traversing trees from root to leaf. More recent work on generative retrieval directly map the query to the item ids in sequence-to-sequence or decoder-only setups [4, 11, 53, 55, 57]. Combinations of these approaches have also been studied, with some performing coarse-grained retrieval with generative approaches, followed by re-ranking using dot products [15]. Finally, the similarity function can be directly parameterized by carefully designed deep neural networks that take various forms [21, 48, 58, 59].

Supporting efficient retrieval with these diverse learned similarities is challenging. Learned similarity functions are generally expensive to compute; with learned index structures, traversing a binary tree with 4 million items requires running beam search for 20 non-parallelizable steps [62], while recommendation and NLP deployments commonly need to handle billions of items [6, 13, 35] with a latency budget of tens of milliseconds. When an arbitrary deep neural network is employed, it's no longer clear how to perform top-$K$ retrieval other than through brute-force [21] or heuristics [59]. While graph-based methods can be used to prune the search space [24, 37, 43, 56], such methods tend to be much slower compared with MIPS algorithms leveraging quantization at high recall rates [1, 19], and their performance can degrade when the similarity function is not a distance metric [39]. What is worse, these algorithms vary significantly in terms of their exact formulations, and the lack of a universal interface makes it even more difficult to design a general solution for efficient retrieval.

Taking a step back, our key insight is that learned similarity approaches are but different ways to increase the expressiveness of the retrieval stage. Formally, for a query $q$ and an item $x$, the expressiveness of the similarity function boils down to deriving alternative parameterizations of $p(x|q)$ matrices, with full rank matrices being the most expressive among them. Dot products, on the other hand,

induces a low-rank bottleneck due to the dimensionality of the embedding, i.e., $\ln p(x|q) \propto \langle f(q), g(x) \rangle$ ($f(q), g(x) \in \mathbb{R}^d$). This cannot be alleviated by simply increasing the embedding dimension $d$, due to memory bandwidth being the main bottleneck in modern dot-product based retrieval systems, such as vector databases [9, 26, 59], and overfitting issues that come with larger embedding dimensions due to the common need to co-train or finetune query- and item-encoders from data [10, 15, 28, 35, 40, 41, 60].

This insight enables us to support efficient retrieval with expressive learned similarity functions by approximating them with MoL. To the best of our knowledge, this is the first work that tackles the problem of efficient retrieval with universal learned similarities, while setting new state-of-the-art results across *heterogeneous* scenarios. We first show that Mixture-of-Logits (MoL) is a universal approximator as it can express $p(x|q)$ matrices of arbitrary high rank, and hence approximate all learned similarity functions (Section 2.1). Our work lays theoretical foundations for MoL's empirical impressive performance gains of 20%-30% across Hit Rate@50-400 on web-scale corpus with hundreds of millions to billions of items [6, 59], and further enables MoL to be effectively applied across diverse retrieval scenarios, from large-scale recommendation systems to finetuning language models for question answering (Section 2.2). We next propose techniques to retrieve the approximate top-$K$ results using MoL with a tight bound (Section 3). Our solution leverages existing widely used APIs of vector databases like top-K queries, thus benefiting from prior work on efficient vector search like MIPS [19, 25, 26, 51]. We empirically compare our techniques with existing approaches, showing that MoL sets new state-of-the-art results on recommendation retrieval and question answering tasks, and our approximate top-k retrieval with learned similarities outperforms baselines by up to 105× in latency, while achieving > .99 recall rate of exact algorithms (Section 4). Importantly, our approach with learned similarities efficiently utilizes modern accelerators due to MoL's higher arithmetic intensity [59], which results in MIPS-level inference latency and throughput. Overall, our work provides strong theoretical and practical justifications to migrate away from the broadly adopted MIPS solution in vector databases to Retrieval with Learned Similarities (RAILS) on GPUs.

## 2 Mixture of Logits

In this section, we describe Mixture of Logits (MoL), propose a load balancing loss to improve conditional computations in MoL, prove that MoL is expressive enough to represent any learned similarity function, and demonstrate how to apply MoL to retrieval tasks. Table 1 summarizes the notations in this paper.

We first describe Mixture of Logits (MoL).

*Mixture of Logits (MoL).* MoL [59] assumes that the query $q$ and the item $x$ are already mapped to $P$ groups of low-rank embeddings ("component-level embeddings"), $f_p(q), g_p(x) \in \mathbb{R}^{d_P}$, where $f_p(q), g_p(x)$ are parameterized with some neural networks based on query and item features, respectively, and $d_P$ is the dimensionality of the low-rank embeddings. MoL then calculates the similarity between the query $q$ and the item $x$ by applying adaptive gating weights, $\pi_p(q, x) \in [0, 1]$, to the inner products of these $P$ pairs of low-rank embeddings, or $\langle f_p(q), g_p(x) \rangle$s. Note that prior work assumes that $\sum_p \pi_p(q, x) = 1$ [6, 59], but this does not affect our analyses in this paper. Following [59]:

$$\phi(q, x) = \sum_{p=1}^{P} \pi_p(q, x) \langle f_p(q), g_p(x) \rangle \qquad (1)$$

To extend this to large-scale datasets and to enable hardware-efficient implementations on accelerators like GPUs, Equation 1 was further modified by decomposing those $P$ dot products as (batched) outer products of $P_q$ query-side and $P_x$ item-side embeddings, where $P_q \times P_x = P$, and applying l2-norm to $f_p(q)$s and $g_p(x)$s:

$$\phi(q, x) = \sum_{p_q=1}^{P_q} \sum_{p_x=1}^{P_x} \pi_{p_q, p_x}(q, x) \left\langle \frac{f_{p_q}(q)}{||f_{p_q}(q)||_2}, \frac{g_{p_x}(x)}{||g_{p_x}(x)||_2} \right\rangle \qquad (2)$$

We use Equation 1 and 2 interchangeably as the MoL form to analyze throughout the rest of this paper, given that the embedding normalization for $f_{p_q}(q)$s and $g_{p_x}(x)$s can be precomputed.

*Mixture of Logits (MoL) with load balancing regularization loss.* We further observe $\pi_p(q, x)$ defines conditional computation to be performed over the $p$ low-rank embedding pairs, or $(f_p(q), g_p(x))$s. $\pi_p(q, x)$ should hence satisfy two conditions:

- Globally, the $p$ low-rank embedding pairs, or $(f_p(q), g_p(x))$s, should receive a similar number of training examples even when $p$ is large and $\pi_p(q, x)$ is sparse, with load distributed evenly across the $p$ pairs. One way to do this is to maximize the entropy $H(p)$ over these embedding pairs.
- The low-rank embedding pairs used to compute each $\phi(q, x)$ should be non-uniform and ideally sparse; e.g., it's desirable to avoid the degenerate solution where $\pi_p(q, x) = \frac{1}{P}$.

| Notation | Description |
|---|---|
| $q$ $(Q, |Q|)$ | query (set of queries, number of queries) |
| $x$ $(X, |X|)$ | item (set of items, number of items) |
| $\phi(q, x)$ | the learned similarity function, i.e., Mixture-of-Logits (MoL). |
| $P$ $(P_q, P_x)$ | MoL uses $P$ low-rank embeddings ("component-level embeddings") to represent $q$ and $x$. With the (batched) outer product form of MoL, $P_q$ and $P_x$ are the numbers of embeddings for $q$ and $x$, respectively; $P = P_q \times P_x$. |
| $\pi_p(q, x)$ $(\pi_{p_q, p_x}(q, x))$ | weight for the $p$-th (or $p_q$-th by $p_x$-th with outer product) embedding set for $(q, x)$. |
| $f(q)$ $(f_p(q))$ | learned embedding for the query ($p$-th component-level query embedding) |
| $g(x)$ $(g_p(x))$ | learned embedding for the item ($p$-th component-level item embedding) |
| $d_P$ | dimensionality of low-rank (component-level) embeddings. $f_p(q), g_p(q) \in \mathbb{R}^{d_P}$. |
| $\langle f(q), g(x) \rangle$ | the dot product similarity function; $\langle f(q), g(x) \rangle = g(x)^T f(q)$. |

**Table 1: Table of Notations.**

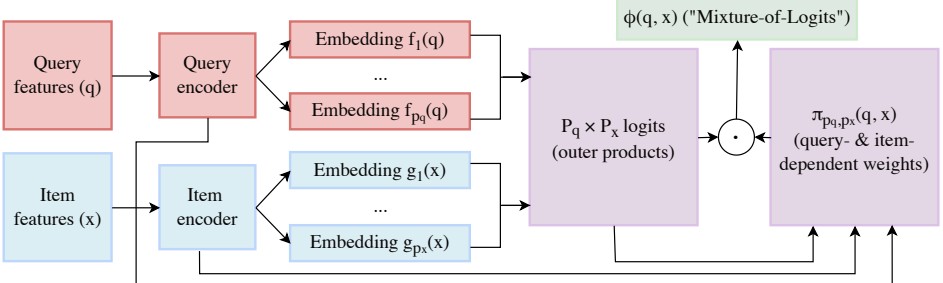

**Figure 1: Mixture-of-logits (MoL) learned similarity.**

One way to do this is to minimize the conditional entropy $H(p|(q, x))$ of $p$ given (query, item) pairs.

Given these two desired conditions, we propose a mutual information-based regularization loss for load balancing, defined as

$$\mathcal{L}_{MI} = -H(p) + H(p|(q, x)) \quad (3)$$

with the overall training loss as

$$-\log \frac{\exp(\phi(q, x))}{\exp(\phi(q, x)) + \sum_{x' \in \mathbb{X}} \exp(\phi(q, x'))} + \alpha \mathcal{L}_{MI} \quad (4)$$

where the first part of Equation 4 is the sampled softmax loss used in [59], and the second part adjusts the weight for the mutual information-based load balancing loss with a hyperparameter $\alpha$.

### 2.1 Expressiveness of Mixture of Logits

Now we show that any high-rank matrix can be decomposed into a mixture of logits based on low-rank matrices, i.e., MoL is a universal approximator. Without loss of generality, we prove the following:

THEOREM 1. *MoL decomposition: Let $A$ be a matrix of $n \times m$, where $n \leq m$. There exists $\pi_1, B_1, \pi_2, B_2, \cdots, \pi_p, B_p$ such that $|A - \sum_{p=1}^{P} \pi_p \circ B_i| < \epsilon$, where $\epsilon$ is a small positive number. Here $B_i$ is a matrix of $n \times m$ with rank equal to or less than $d$, and $\pi_1, \pi_2, \cdots, \pi_P$ are $n \times m$ matrices that together define a probability distribution over each $(i, j)$ tuple, such that $\sum_{p=1}^{P} \pi_p(i, j) = 1, 0 \leq \pi_p(i, j) \leq 1$ for any $1 \leq i \leq n, 1 \leq j \leq m, 1 \leq p \leq P$.*

We can think about $n$ as the number of queries and $m$ the number of items (or vice versa). First, the theorem trivially holds if the rank of $A$ is less than or equal to $d$ ($d \leq n$):

LEMMA 1. *MoL decomposition when $Rank(A) \leq d$: Let $A$ be a matrix as defined in Theorem 1. If the rank of $A$ is less than or equal to $d$, then we have $A = \pi \circ A$, where $\pi(i, j) = 1$ for any $1 \leq i \leq n, 1 \leq j \leq m$.*

Then we prove for the case where the rank of $A$ is greater than $d$. Without loss of generality, we prove the case where the matrix has full rank, i.e., $Rank(A) = n$:

LEMMA 2. *MoL decomposition when $Rank(A) = n$: Let $A$ be a matrix as defined in Theorem 1. Then there exists $\pi, B_1, B_2$ such that $|A - (\pi \circ B_1 + (1 - \pi) \circ B_2)| < \epsilon$, where $Rank(B_1) \leq d, Rank(B_2) \leq d$, and $0 \leq \pi(i, j) \leq 1$ for $1 \leq i \leq n, 1 \leq j \leq m$.*

PROOF. Because $A$ is a matrix of rank $n$, it can be rewritten as $A = UI_nV$, where $I_n$ is an identity matrix with rank $n$. Thus, $A_{ij} = \sum_{k=1}^{n} U_{ik}V_{kj}, 1 \leq i \leq n, 1 \leq j \leq m$. Let $A'$ be a matrix of

$n \times m$, where $A'_{ij} = \lambda_{ij} \cdot \sum_{k=1}^{d} U_{ik}V_{kj}$ for $1 \leq i \leq n, 1 \leq j \leq m$. Here, $\lambda_{ij} = 1 + \frac{\sum_{k=d+1}^{n} U_{ik}V_{kj}}{\sum_{k=1}^{d} U_{ik}V_{kj}}$ if $\sum_{k=1}^{d} U_{ik}V_{kj} \neq 0$, otherwise $\lambda_{ij} = 1 + \frac{\sum_{k=d+1}^{n} U_{ik}V_{kj}}{\epsilon}$. Thus, we have $|A - A'| \leq \epsilon$.

Let $\lambda_{min} = \min \lambda_{ij}$, and $\lambda_{max} = \max \lambda_{ij}$. Let $B_1 = \lambda_{min}UD_{n,d}V$, $B_2 = \lambda_{max}UD_{n,d}V$, where $D_{n,d}$ denotes an $n$-by-$n$ diagonal matrix with the first $d$ elements of the diagonal being 1s and the rest being 0s. We have $A'_{ij} = \lambda_{ij} \sum_{k=1}^{d} U_{ik}V_{kj} = \pi(i, j) \cdot B_{1ij} + (1 - \pi(i, j)) \cdot B_{2ij}$, where $\pi(i, j) = \frac{\lambda_{max} - \lambda_{ij}}{\lambda_{max} - \lambda_{min}}$. Because $\lambda_{min} \leq \lambda_{ij} \leq \lambda_{max}$, we have $0 \leq \pi(i, j) \leq 1$.

Thus, we have constructed $B_1, B_2, \pi$ such that $|A - (\pi \circ B_1 + (1 - \pi) \circ B_2)| = |A - A'| \leq \epsilon$. □

*Remark* Here, we have shown that any high-rank matrix can be expressed as a mixture of logits of two low-rank matrices. Note that our decomposition is not intended to be used as a distillation of the original high-rank matrix. It is likely prohibitively expensive to populate the full matrix with a learned similarity function. In addition, our proof also does not indicate that having two mixture components is sufficient to train the embeddings and the learned similarity function. It is well-known that overparameterization is often necessary to enable efficient and performant training.

### 2.2 Applying MoL to Heterogeneous Use Cases

We now discuss how to apply MoL to retrieval tasks in different domains. Parameterization of the low-rank, component-level embeddings, or $f_p(q), g_p(x) \in \mathbb{R}^{d_P}$, plays an important role in realizing MoL's theoretical expressiveness in practice, as suggested by prior work [6]. We discuss two scenarios on the opposite end of the spectrum, one with *a large number of heterogeneous features* – retrieval in large-scale recommendation systems, followed by another with *a single homogeneous feature* – finetuning language models for question answering and related NLP use cases, shown in Figure 2.

*Retrieval in Large-scale Recommendation Systems.* Recommendation systems are characterized by the large number of heterogeneous features they use [10, 52, 60]. This naturally enables some of those features to be utilized on the query- (user-) or on the item-side. For instance, embeddings can be constructed based on cluster ids on both the query-side and the item-side [6]. For common benchmark datasets, User ID-based one-hot embeddings [30] represent another possible $g_p(q)$ to use, which we evaluate in Section 4.

*Finetuning Language Models for Question Answering.* In contrast, language models are characterized by their use of homogeneous

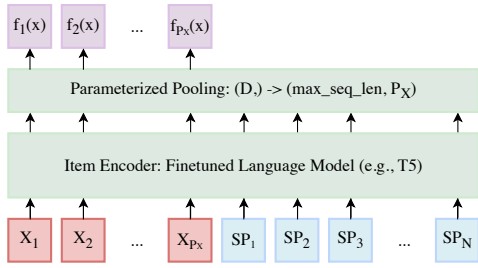
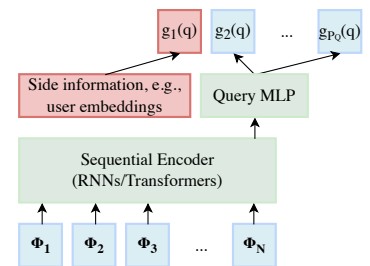

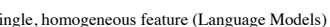

Single, homogeneous feature (Language Models)               Rich, heterogeneous features (Recommendations)

**Figure 2: Illustration of how to apply Mixture-of-logits (MoL) learned similarity to various retrieval scenarios, with a language model finetuning use case (characterized by a single homogeneous feature) shown on the left, and a recommendation use case (characterized by a large number of heterogeneous features) shown on the right. More details can be found in Appendix A.2.**

semantic features, such as wordpieces and sentencepieces [31]. We observe that MoL can be similarly adopted for those use cases. To obtain the $P_X$ item embeddings for MoL, we expand tokenizer's vocabulary with $P_X$ special aggregation tokens $X_1, \ldots, X_{P_X}$, and append those $P_X$ tokens at the beginning of every tokenized sequence, $SP_1, \ldots, SP_N$, as illustrated in Figure 2 [1]. These $P_X$ special tokens play similar roles as the CLS token in BERT [12], and during finetuning of the language model, are co-trained to aggregate different aspects of information as inputs for MoL. Additionally, we can design a learned pooling function to adapt pooling policy at an example-level ("Parameterized Pooling") to improve model quality, which we discuss further in Appendix A.2.

## 3 Retrieval Algorithms

In this section, we describe the problem of retrieving the top $K$ items with *MoL* as well as exact and approximate retrieval algorithms. Formally, we define the top $K$ retrieval problem as the following:

**DEFINITION 1.** ***Top $K$ with MoL***: *Let $q$ be a query and $X$ be a set of items, where both the query $q$ and each item $x \in X$ are associated with $P$ embeddings. Together we have $P$ pairs of embeddings, $(f_p(q), g_p(x)), 1 \le p \le P$. Let $\phi(q, x) = \sum_{p=1}^{P} \pi_p(q, x) \langle f_p(q), g_p(x) \rangle$ be the similarity score of $q, x$, where $x \in X$. The top $K$ query with MoL returns the $K$ items from $X$ with the highest $\phi(q, x)$s.*

For approximate top $K$ retrieval with *MoL*, we define the gap of the approximate and exact top $K$ results as follows:

**DEFINITION 2.** ***Gap of approximate top $K$:*** *Let $q$ be a query and $X_K$ be the set of exact top $K$ items for the query $q$ from a set of items $X$. Let $X^*$ be the approximate top $K$ results, where $X^* \subseteq X$. Let $S = \min\{\phi(q, x), x \in X^*\}$ and $S' = \max\{\phi(q, x), x \in X_K \setminus X^*\}$. We call $S_\Delta = S' - S$ the gap of the top $K$ with $X^*$.*

### 3.1 Exact algorithm

The brute-force algorithm to retrieve the exact top $K$ with *MoL* is to evaluate $\phi(q, x)$ for each query $q$ and item $x$. This algorithm can be prohibitively expensive if the number of items is large. Instead, we describe a more efficient two-pass algorithm to retrieve the exact top $K$ items as shown in Algorithm 1.

---

[1]Note that many question answering scenarios [11, 28, 41, 53, 57] utilize bidirectional language models for retrieval, like BERT [12] or T5 [44]; for recent unidirectional language models, we can add $X_1, \ldots, X_{P_X}$ to the end of the input sequence instead.

---

**Algorithm 1** Exact top $K$ algorithm.

---

**Input:** query $q$, a set of items $X$, $f_p(\cdot)$, $g_p(\cdot)$ for constructing the component-level embeddings $f_p(q), g_p(x)$

**Output:** exact top $K$ items

1: $G \leftarrow \emptyset$
2: **for** $p \in P$ **do**
3:   $X_p \leftarrow \{g_p(x), x \in X\}$      ▷ Can be preprocessed.
4:   $G \leftarrow G \cup TopKDotProduct(f_p(q), X_p)$   ▷ Retrieve top $K$ items
  for each pair of embeddings
5: $S_{min} \leftarrow \infty$
6: **for** $x \in G$ **do**
7:   $s \leftarrow MoL(q, x)$
8:   **if** $s < S_{min}$ **then** $S_{min} \leftarrow s$
9: $G' \leftarrow \emptyset$
10: **for** $p \in P$ **do**
11:   $G' \leftarrow G' \cup RangeDotProduct(f_p(q), S_{min}, X_p)$   ▷ Retrieve all
  items $x \in X_p$ with $\langle f_p(q), x \rangle \ge S_{min}$.
12: **return** $BruteForceTopKMoL(q, G')$▷ Retrieve the top $K$ items from
  $G'$ with $MoL$

---

We start by retrieving the top $K$ items with the highest dot product scores for each group of embeddings as the initial candidate set $G$ (line 1-4). Then we evaluate the *MoL* scores of the items in $G$ and find the minimal learned similarity score $S_{min}$ (line 5-8). Next we retrieve all items within a distance of $S_{min}$ with the query $q$ as the candidate set $G'$ (line 9-11). Finally, we evaluate the *MoL* scores of the items in $G'$, and return the top $K$ items with the highest scores (line 12).

We argue that Algorithm 1 retrieves the exact top $K$ items with *MoL*. Let $X_K$ be the set of the exact top $K$ items and $X'$ be the result of Algorithm 1. Let $x \in X_K$ and $\phi(q, x)$ be the *MoL* score of $x$ and $q$. Since $x$ has the highest top $K$ score with *MoL*, $\phi(q, x) \ge S_{min}$. Since the *MoL* score is a weighted score over the dot product scores, we have $\max\{\langle f_p(q), g_p(x) \rangle, 1 \le p \le P\} \ge \phi(q, x) \ge S_{min}$. Since Algorithm 1 retrieves all the items with a dot product higher than or equal to $S_{min}$ of $q$ for each embedding $q_p$ (line 9-11), we have $x \in G'$. Thus, $x \in X'$. So we have shown that $X_K = X'$.

### 3.2 Approximate algorithms

In the exact algorithm shown in Algorithm 1, we need to retrieve all the items with a dot product higher than or equal to a threshold. When the threshold is a loose filter of the item set, which may happen when the dot product scores are skewed, $G'$ can be large,

and the evaluation of *MoL* over a large number of candidate items can still be expensive. Here, we describe two heuristics to approximately retrieve the top $K$ items and analyze their gap against the exact top $K$ algorithm.

In both heuristics, we perform a two-stage retrieval as shown in Algorithm 2. In the first stage, we retrieve a set of $K'$ candidate items that are potentially high in *MoL* score by using dot products (line 2). Note that $K'$ can be larger than $K$, e.g., due to oversampling. In the second stage, we evaluate the *MoL* scores of the candidate items and return the top $K$ items (line 3).

---

**Algorithm 2** Approximate top $K$ algorithms.

---

**Input:** a query $q$, a set of items $X$
**Output:** approximate top $K$ items
1: **function** ApproxTopK$(q, X, K, K')$
2:     $G \leftarrow TopKCandidate(q, X, K')$ ▷ Retrieve the top $K'$ candidates
3:     **return** $BruteForceTopKMoL(q, G, K)$     ▷ Retrieve the top $K$ items with *MoL*
**Input:** a query $q$, a set of items $X$, $f_p(\cdot)$, $g_p(\cdot)$ for constructing the component-level embeddings $f_p(q), g_p(x)$
**Output:** union of top $K$ items over $P$ component-level embeddings by dot product
4: **function** TopKPerEmbedding$(q, X, K)$
5:     $G \leftarrow \emptyset$
6:     **for** $p \in P$ **do**
7:        $X_p \leftarrow \{g_p(x), x \in X\}$     ▷ Can be preprocessed.
8:        $G \leftarrow G \cup TopKDotProduct(f_p(q), X_p, K)$ ▷ Retrieve the top $K$ items by dot product
9:     **return** $G$
**Input:** a query $q$, a set of items $X$, $f_p(\cdot)$, $g_p(\cdot)$ for constructing the component-level embedding $f_p(q), g_p(x)$
**Output:** top $K$ items based on the averaged dot product, $\sum_p \langle f_p(q), g_p(x) \rangle / P$.
10: **function** TopKAvg$(q, X, K)$
11:     $q' \leftarrow \sum_{p=1}^{P} f_p(q)$
12:     $X' \leftarrow \{\sum_{p=1}^{P} g_p(x)/P, x \in X\}$     ▷ Can be preprocessed.
13:     **return** $TopKDotProduct(q', X', K)$

---

Here, we describe two heuristics to retrieve the candidate items:

*Top $K$ per embedding.* Given a query $q$ and a set of items $X$, for each embedding set $p$, retrieve top $K$ items $X_{K,p}$ based on dot product ($\langle f_p(q), g_p(x) \rangle$). Return the union across $P$ queries.

The top $K$ per embedding heuristic returns the union of the top $K$ items for each embedding by dot product. We analyze the gap of this heuristic as follows:

THEOREM 2. *Upper bound of the gap of top $K$ **per embedding:** Let $X_{K,p}$ be the top $K$ items of the embedding set $p$ and $S = \max\{\phi(q, x), x \in X_{K+1,p}\}$. Let $S_{min}$ be the $K^{th}$ largest MoL score of the items in $\cup_p X_{K,p}$, then the gap of $S_\Delta \leq S' - S_{min}$.*

*Remark* Note that there exists an *MoL* such that $S_\Delta = S - S_{min}$, i.e., when $\pi_p(q, x) = 1$ for $x_p = \arg\max_{x,p}\{\langle f_p(q), g_p(x) \rangle, x \in X_{K+1,p} \setminus X_{K,p}\}$. Thus, the upper bound of $S_\Delta$ is tight.

*Top $K$ average.* Given a query $q$ and a set of items $X$, return the top $K$ items with the highest average dot product $\sum_p \langle f_p(q), g_p(x) \rangle / P$.

Note that the top $K$ average heuristic returns the exact top $K$ items when the gating weight distribution in *MoL*, $\pi$, is uniform.

This heuristic is interesting for two reasons. First, the items retrieved by this heuristic are likely to be the top $K$ items of *MoL* when the weight distribution is more balanced. This complements the heuristic that retrieves top $K$ per embedding. Second, in the setup where the set of embedding pairs is constructed as the outer product of the embeddings of a query and those of an item (Equation 2), the average dot product can be efficiently preprocessed and materialized for the items, and the computation of the top $K$ average is then *agnostic* to the number of embedding pairs, $P = P_q \times P_x$.

Formally, let $P = P_q \cdot P_x$ be the number of embedding pairs, where $P_q$ is the number of embeddings of a query $q$ and $P_x$ is that of an item $x$. The average dot product can be computed as

$$\frac{1}{P} \cdot \sum_{p=1}^{P} \langle f_p(q), g_p(x) \rangle = \frac{1}{P} \cdot \sum_{p_q=1}^{P_q} \sum_{p_x=1}^{P_x} \langle f_{p_q}(q), g_{p_x}(x) \rangle \quad (5)$$

$$= \frac{1}{P} \cdot \left\langle \sum_{p_q=1}^{P_q} f_{p_q}(q), \sum_{p_x=1}^{P_x} g_{p_x}(x) \right\rangle \quad (6)$$

Thus, we can preprocess the embeddings of the items and the query, so the number of embeddings accessed is 1 per item for a given query, regardless of the overall number of component-level embeddings used by *MoL*, i.e., $P$.

Finally, we can combine the candidates retrieved from top $K$ per embedding group and the top $K$ average as the following:

*Combined top $K$.* Given a query $q$, a set of items $X$, and $K$, return the union of the items from the top $K$ per embedding group across the $P$ groups and the top $K$ items from the top $K$ average.

THEOREM 3. *Upper bound of the gap of combined top $K$. Let $X_{K,p}$ be the top $K$ items of the embedding set $p$ and $S_{min}$ as defined in Theorem 2. Let $X'_K$ be the top $K$ items from top $K$ average. Let $S' = \max\{\phi(q, x), x \in X \setminus (\cup_p X_{K,p} \cup X'_K)\}$. Then the gap of $S_\Delta \leq S' - S_{min}$.*

*Remark* Similar to Theorem 2, the upper bound of the gap is tight. In practice, we can configure the $K$ to be different for the two heuristics, i.e., $K_1$ and $K_2$. For example, when the weight distribution $\pi$ is more balanced, $K_2$ can be configured to be larger as the top $K$ average approach approximates *MoL* well while being more computationally efficient.

## 4 Evaluation

In this section, we evaluate the performance of the *MoL* based learned similarity with the proposed load balancing loss, and the efficiency of our retrieval algorithms discussed in Section 3. Our code and model checkpoints are available at the following anonymized GitHub repository: https://anonymous.4open.science/r/rails-4E62.

### 4.1 Workloads

We benchmark MoL with the proposed load balancing loss $\mathcal{L}_{MI}$, on top of state-of-the-art baselines in recommendation systems and question answering. We describe workloads used below.

*Recommendation Systems.* We consider three widely used datasets, the 1M and 20M subsets of MovieLens [20], and the largest Books subset of Amazon Reviews [38]. Sequential retrieval models have been shown to achieve state-of-the-art results on these datasets [22,

27, 60]. In these settings, sequential encoders, like RNNs or Transformers, are used to map user representations at time $t$ – e.g., in a commonly used setting shown in Figure 2, the list of items in user history up until time $t$, $\Phi_0, \ldots, \Phi_t$ – to $\mathbb{R}^d$, and the model is trained to autoregressively predict the next item $x_{t+1}$. We hence compare MoL with the proposed regularization loss on top of two popular backbones used for sequential retrieval models, SASRec [27] and HSTU [60], against cosine similarity baselines. We utilize user id-based embeddings discussed in Section 2.2 and MLPs to parameterize the $P_Q$ query-side and the $P_X$ item-side features.

*Question Answering (QA).* Natural Questions (NQ) [32] is commonly used to evaluate state-of-the-art neural retrieval models, including dense retrieval [28, 41] and generative retrieval [11, 53, 55, 57] approaches in recent years. The most commonly used version [53, 55, 57], which we reuse in our work, is often referred to as NQ320k. NQ320k consists of 320k query-items pairs, where the items are from Wikipedia pages and the queries are natural language questions. We utilize special aggregation tokens discussed in Section 2.2 to parameterize embeddings in MoL, and compare MoL with popular sparse retrieval methods [42, 47], dense retrieval methods [28, 40, 41], and generative retrieval methods [4, 11, 53, 55, 57]. Consistent with recent work [53, 57, 63], we use the pre-trained query generation model from DocT5Query [42] to generate synthetic (query, item) pairs for data augmentation.

Table 2 summarizes the statistics of these four workloads.

## 4.2 Quality of MoL-based Learned Similarity

*Metrics.* We use Recall (Hit Rate) as the main metric. We report Hit Rate@{1, 10, 100} and Mean Reciprocal Rank (MRR) on NQ320K, following [53, 57], and Hit Rate@{1, 10, 50, 200} on *ML-1M*, *ML-20M*, and *Books*, following [59, 60].

*Hyperparameter Settings.* We set the weight $\alpha$ for the proposed load balancing loss $\mathcal{L}_{MI}$ to 0.001 for all experiments. We reuse baseline settings for most other hyperparameters, including learning rate, number of examples used for in-batch negative sampling, etc., with detailed discussions in Appendix A. For the NQ320K dataset, we reuse SEAL [4] and NCI [57] results reported by [57], and results for other models as reported by [53]. The Sentence-T5 [40], GENRE [11], DSI [55], SEAL [4], DSI+QG [63], NCI [57], and GenRet [53] rows are all finetuned from T5-base, consistent with MoL, to ensure a fair comparison. All other results are reimplemented ourselves in PyTorch, and are trained with 1x/2x 48GB GPUs for the recommendation datasets and 4x 80GB GPUs for the QA datasets.

*Results.* Across the six recommendation scenarios utilizing different sequential encoder backbones, Mixture-of-Logits (MoL rows) consistently outperform dot products by an average of 18.5% in MRR, 22.0% in HR@1, and 18.5% in HR@10 (Table 3). On the widely

| Workload | $\|Q\|$ | $\|X\|$ | $\|P_q\|$ | $\|P_x\|$ | $d_P$ |
|---|---|---|---|---|---|
| *ML-1M* | 6,040 | 3,649 | 8 | 4 | 64 |
| *ML-20M* | 138,493 | 24,186 | 8 | 4 | 128 |
| *Books* | 694,897 | 674,044 | 8 | 8 | 32 |
| NQ320K | 307,373 | 109,739 | 4 | 4 | 768 |

**Table 2: Workload statistics.**

| Method | HR@K | | | | MRR |
|---|---|---|---|---|---|
| | K=1 | K=10 | K=50 | K=200 | |
| *ML-1M dataset* | | | | | |
| SASRec [27] | .0610 | .2818 | .5470 | .7540 | .1352 |
| SASRec + MoL | .0697 | .3036 | .5617 | .7667 | .1441 |
| HSTU [60] | .0750 | .3332 | .5956 | .7824 | .1579 |
| HSTU + MoL | **.0884** | **.3465** | **.6022** | .7935 | **.1712** |
| HSTU + MoL abl. $\mathcal{L}_{MI}$ | .0847 | .3417 | .6011 | **.7942** | .1662 |
| *ML-20M dataset* | | | | | |
| SASRec [27] | .0653 | .2883 | .5484 | .7658 | .1375 |
| SASRec +MoL | .0778 | .3102 | .5682 | .7779 | .1535 |
| HSTU [60] | .0962 | .3557 | .6146 | .8080 | .1800 |
| HSTU + MoL | **.1010** | **.3698** | **.6260** | **.8132** | **.1881** |
| HSTU + MoL abl. $\mathcal{L}_{MI}$ | .0994 | .3670 | .6241 | .8128 | .1866 |
| *Books dataset* | | | | | |
| SASRec [27] | .0058 | .0306 | .0754 | .1431 | .0153 |
| SASRec + MoL | .0095 | .0429 | .0915 | .1635 | .0212 |
| HSTU [60] | .0101 | .0469 | .1066 | .1876 | .0233 |
| HSTU + MoL | **.0156** | **.0693** | **.1362** | .2144 | **.0329** |
| HSTU + MoL abl. $\mathcal{L}_{MI}$ | .0139 | .0661 | .1315 | **.2153** | .0323 |

**Table 3: Evaluation of performance for sequential retrieval models on MovieLens and Amazon Reviews.**

| Method | HR@K | | | MRR |
|---|---|---|---|---|
| | K=1 | K=10 | K=100 | |
| *Sparse retrieval* | | | | |
| BM25 [47] | .297 | .603 | .821 | .402 |
| DocT5Query [42] | .380 | .693 | .861 | .489 |
| *Dense retrieval* | | | | |
| DPR [28] | .502 | .777 | .909 | .599 |
| Sentence-T5 [40] | .536 | .830 | .938 | .641 |
| GTR-Base [41] | .560 | .844 | .937 | .662 |
| *Generative retrieval* | | | | |
| GENRE [11] | .552 | .673 | .754 | .599 |
| DSI [55] | .552 | .674 | .780 | .596 |
| SEAL [4] | .570 | .800 | .914 | .655 |
| DSI+QG [63] | .631 | .807 | .880 | .695 |
| NCI [57] | .659 | .852 | .924 | .731 |
| GenRet [53] | .681 | .888 | .952 | .759 |
| *Learned similarities* | | | | |
| MoL | **.685** | **.919** | **.970** | **.773** |
| MoL abl. $\mathcal{L}_{MI}$ | .673 | **.919** | .968 | .767 |

**Table 4: Evaluation of performance for QA retrieval models finetuned from language models on Natural Questions.**

used Natural Questions QA dataset, MoL outperforms all recent generative retrieval approaches as well as strong dot product (dense retrieval) baselines (Table 4). These results validate that learned similarities, in particular MoL, are not only theoretically expressive but also *practically learnable*, improving retrieval quality across heterogeneous scenarios, including sequential retrieval models for Recommendations and finetuning LMs for Question Answering.

*Ablation Studies.* We conduct ablation studies for the proposed mutual information-based load balancing loss relative to the best performing method for each dataset ("abl. $\mathcal{L}_{MI}$" rows). Results show that our proposed $\mathcal{L}_{MI}$ loss improves HR@1 by 4.6%, HR@10

| | Method | HR@1 | HR@5 | HR@10 | HR@50 | HR@100 | Latency/$ms$ |
|---|---|---|---|---|---|---|---|
| | *BruteForce* | 1.00 | 1.00 | 1.00 | 1.00 | 1.00 | 3.17±.03 |
| | *TopKPerEmbd*5 | .762 | .707 | .647 | .468 | .402 | 1.28±.04 |
| | *TopKPerEmbd*10 | .956 | .881 | .820 | .646 | .564 | 1.31±.04 |
| *ML-20M* | *TopKPerEmbd*50 | **1.00** | **.992** | .982 | .933 | .900 | 1.63±.02 |
| | *TopKPerEmbd*100 | **1.00** | **1.00** | **.999** | .981 | .966 | 2.41±.04 |
| | *TopKAvg*200 | **1.00** | **.998** | **.997** | .982 | .959 | .86±.04 |
| | *TopKAvg*500 | **1.00** | **1.00** | **1.00** | **1.00** | **.998** | .88±.03 |
| | *CombTopK*5_200 | **1.00** | **1.00** | **1.00** | **1.00** | **.998** | 1.46±.04 |
| | *BruteForce* | 1.00 | 1.00 | 1.00 | 1.00 | 1.00 | 181.34±9.09 |
| | *TopKPerEmbd*5 | .907 | .915 | .809 | .509 | .396 | 26.40±.63 |
| | *TopKPerEmbd*50 | **.993** | **.992** | **.994** | .956 | .902 | 27.59±.68 |
| | *TopKPerEmbd*100 | **1.00** | **.995** | **.996** | **.985** | .959 | 29.64±.65 |
| | *TopKAvg*200 | **1.00** | .978 | .948 | .845 | .767 | 0.81±.11 |
| | *TopKAvg*500 | **1.00** | **.992** | **.996** | .919 | .875 | 0.80±.09 |
| *Books* | *TopKAvg*1000 | **1.00** | **1.00** | **.996** | .963 | .939 | 0.87±.04 |
| | *TopKAvg*2000 | **1.00** | **1.00** | **1.00** | .980 | .968 | 1.11±.04 |
| | *TopKAvg*4000 | **1.00** | **1.00** | **1.00** | **.996** | **.987** | 1.72±.04 |
| | *CombTopK*5_200 | .979 | .984 | .981 | .892 | .818 | 25.73±.76 |
| | *CombTopK*50_500 | **.993** | **.989** | **.994** | .975 | .961 | 27.99±.65 |
| | *CombTopK*100_1000 | **1.00** | **.997** | **.996** | **.993** | **.992** | 30.40±.67 |
| | *BruteForce* | 1.00 | 1.00 | 1.00 | 1.00 | 1.00 | 37.74±.47 |
| | *TopKPerEmbd*5 | **1.00** | **1.00** | **.995** | .961 | **1.00** | 4.71±.08 |
| | *TopKPerEmbd*10 | **1.00** | **1.00** | **1.00** | .981 | **1.00** | 4.83±.08 |
| *NQ320K* | *TopKPerEmbd*50 | **1.00** | **1.00** | **1.00** | **1.00** | **1.00** | 6.31±.09 |
| | *TopKAvg*100 | **.999** | **.999** | **.999** | **.998** | **.995** | .57±.05 |
| | *TopKAvg*200 | **1.00** | **1.00** | **1.00** | **.999** | **.999** | .66±.01 |
| | *CombTopK*5_100 | **1.00** | **1.00** | **1.00** | **.999** | **.999** | 5.28±.08 |

**Table 5: Evaluation of top $K$ retrieval performance, with hit rate (HR) normalized by the brute-force top $K$ method and latency with standard deviation (i.e., ±) measured over a batch of queries (where the batch size is 32). (Relative) hit rate higher than .99 is marked in bold.**

by 1.7% and MRR by 1.6% across the four datasets. In particular, our proposed $\mathcal{L}_{MI}$ loss enables MoL to outperform the best generative retrieval approach on NQ320K, GenRet [53], across all metrics.

## 4.3 Top $K$ retrieval performance

We evaluate the following methods for top $K$ retrieval performance:

- Brute-force top $K$ (*BruteForce*): Evaluate the *MoL* scores for all items and return the top $K$ items. This is the ground truth in our top $K$ evaluation [2].
- Per embedding top $K$ (*TopKPerEmbd*($N$)): This algorithm is described in Section 3.2. $N$ is the number of candidate items retrieved from each embedding set, where $N \times P \geq K$.
- Average top $K$ (*TopKAvg*($N$)): This algorithm is described in Section 3.2. $N$ is the number of the candidate items retrieved by average dot products, where $N \geq K$.
- Combined top $K$ from per embedding top $K$ and average top $K$ (*CombTopK*$N_1$_$N_2$): This is described in Section 3.2. $N_1$ is the number of candidate items retrieved from per embedding top $K$ and $N_2$ is the number of candidate items retrieved from average top $K$, where $N_1 \times P + N_2 \geq K$.

---

[2]We omit the baseline with the two-pass exact algorithm (Section 3.1) because the range-based item retrieval can still be expensive when the range threshold is loose. Empirically, the brute-force top $K$ is more efficient on our datasets. We leave the efficient implementation of the two-pass exact algorithm as future work.

For each dataset, we evaluate top $K$ retrieval methods based on the best performing model configurations reported in Table 3 and Table 4. Table 5 shows the hit rate (HR) and latency of all the methods. The hit rate is normalized by the ground truth, i.e., the hit rate achieved with brute-force top $K$. We measure latency by evaluating a batch of 32 retrieval queries, in order to achieve high accelerator utilization; this is consistent with prior work on GPU/TPU-based retrieval algorithms [9, 26, 59]. We omit *ML-1M* as its size is small (Table 2). We perform evaluation on a single RTX 6000 Ada GPU. We report latency averaged over 20 warm runs.

We observe that our approximate heuristics achieve high HR with oversampling. For example, *TopKAvg*500 is > .99 in relative HR across the board for *ML-20M*, and *TopKAvg*100 is > .99 in relative HR across the board for *NQ320K*. In addition, the combined top $K$ algorithm can outperform both *TopKPerEmbd* and *TopKAvg* of the corresponding configurations, sometimes significantly, e.g., *CombTopK*5_200 vs. *TopKPerEmbd*5 and *TopKAvg*200 on *Books*. This indicates that the set of candidate items retrieved by each individual approximate algorithm indeed complements each other when the weight distributions, $\pi_p(q, x)$s, vary in *MoL*.

In terms of efficiency, we observe that our approximate heuristics are significantly lower in latency than the exact baseline, especially as the number of items in the dataset becomes large. For example, compared to *BruteForce*, *TopKAvg* achieves > .99 relative HR@100 with a speedup of 105× and 66× in latency for *Books* and *NQ320K*,

respectively. While the algorithm latency grows with the size of the dataset in the brute-force baseline, it grows much slower with the approximate methods. For example, the algorithm latency increases by 57× from *ML-20M* to *Books* in *BruteForce*, while the growth rate is 12× and 1.0× for *TopKPerEmbd*100 and *TopKAvg*500, respectively. Thus, we expect that the speedup of the approximate methods to become even more prominent with larger datasets.

We also notice that *TopKAvg* tends to be more efficient than *TopKPerEmbd* with comparable HR, e.g., *TopKAvg*2000 vs. *TopKPerEmbd*100 on *Books* with 27× speedup in latency. We believe that this is mainly due to two reasons. First, when the HR is comparable, the maximal number of candidate items from *TopKPerEmbd* is larger than that of *TopKAvg*. Second, compared to *TopKPerEmbd*, the computation of *TopKAvg* is agnostic to the number of component-level low-rank embeddings, $P$, because of the materialization optimization described in Section 3.2. Interestingly, we also see that the combined top $K$ is more efficient than the summation of the latency of its individual components, e.g., *CombTopK*5_200 is 1.5× faster than the sum of the latency from *TopKPerEmbd*5 and *TopKAvg*200 on *ML-20M*. This is because our implementation reduces the overhead of the combined method by consolidating processing shared by the two components.

Overall, empirically *TopKAvg* strikes a good balance between high HR and low latency, and the combined top $K$ algorithm can be used if the target HR is extremely stringent.

## 5 Related work

*Similarity Functions in Retrieval.* Most information retrieval models in recommendation systems and natural language processing (e.g., question answering) follow a classical two-stage paradigm [10, 28], where up to billions of items [6, 13, 35, 59] are first filtered down to hundreds in the *retrieval* stage, followed by another stage (e.g., ranking in recommendation systems or generation in RAG [33]) that produces the final results. Earlier work on large-scale neural retrieval models primarily utilize dual-encoder (dense retrieval, etc.) setups, with dot products as the similarity function [10, 28, 40, 41]. Researchers quickly realized that dot products limited retrieval stage's performance, and explored various learned similarity-based approaches. Prominent variants include maximum similarity based on multiple embeddings [29, 35, 48], specialized neural networks, often leveraging Hadamard products [6, 21, 54, 56], and representing item ids as token sequences ("learned index structures"), either implicitly defined during tree traversal [23, 62, 64] or explicitly in the "generative retrieval" setups [4, 11, 53, 55, 57, 63]. It has been shown, however, that learned neural distances often fail to outperform dot products, e.g., Hadamard MLPs in recommendation systems [46] and DSI for QA scenarios in NLP [53]. Learned index structures further introduce stability and latency challenges as both NLP and recommendation systems need to support billion-scale realtime updated set of items [6, 13, 59]. Despite these challenges, significant gains (17% gains at Hit Rate@100 [59] to 24% gains at Hit Rate@400 [6]) with learned similarities have been reported in recent years; these can be attributed to careful construction of learned similarity functions [48, 59], implicit diversification done as part of beam search [15], explicit incorporation of side-information using special neural architectures [6], and hardware-aware similarity function and inference algorithm design on GPUs [6, 9, 43, 59].

*Load Balancing for Conditional Computations in Neural Networks.* Conditional computations have been widely utilized in deep learning models [2, 8, 49]. Regularization losses have been proposed based on the observation that an ideal policy should evenly utilize all compute units in aggregate while being sparse at an individual example level [2]. Mixture-of-experts, a common way to implement conditional computations, has been widely used in language and vision domains [8, 49] where mutual information-based regularization losses between experts and tasks [8] and experts and tokens [50] have been shown to help with various architectures.

*Efficient Nearest Neighbor Search (NNS).* Nearest neighbor search has been a popular topic of research due to their critical role in large-scale retrieval and vector databases. Most studies focus on the dot product case, also known as Maximum Inner Product Search (MIPS). Various techniques were proposed and analyzed, including tree structures [3, 45], locality sensitive hashing [17, 51], production quantization [18, 25], data partitioning [34, 61], graph-based methods [24, 37], and so on. The general case for NNS utilizing learned similarities remains less studied; for learned index structures, techniques to construct trees have been proposed to ensure beam search result in globally optimal top-$K$ results [64]. Algorithms based on implicit [24, 37, 43, 56] or explicit graphs [56] have been proposed to obtain a tractable candidate set in multi-stage retrieval setups; however, such approaches' performance can degrade when the similarity function is not a metric, and constructing appropriate graph indices for non-metric similarity functions can remain challenging even for the inner product case [39]. Due to GPUs and other accelerators having orders of magnitude higher arithmetic intensity vs CPUs, traditional quantization techniques [18, 51] no longer fully utilize GPUs; accelerator-specific nearest neighbor algorithms that benefit from increased compute have been proposed recently [6, 9, 43, 59].

## 6 Conclusion

We have analyzed techniques for efficient retrieval with expressive learned similarities in this work. We begin by showing Mixture-of-Logits (*MoL*) is a universal approximator of learned similarity functions, and further empirically learnable – *MoL* with our proposed load balancing loss consistently outperforms dot products (dense retrieval), sparse retrieval, and generative retrieval approaches across Recommendation Systems and Question Answering scenarios, setting new state-of-the-art across common, heterogeneous benchmark datasets. We next propose both exact and approximate algorithms to enable efficient retrieval using learned similarity functions, and show their correctness and bounds. Across all datasets evaluated, we demonstrate that our approximate top $K$ algorithms can reach .99 of Hit Rate relative to exact algorithms, while achieving up to 105× reduction in end-to-end latency and with minimal indexing overheads. We expect the speedups to be further amplified with larger-scale datasets and GPU kernel optimizations. Given MoL's empirical impressive performance gains of 20%-30% across Hit Rate@50-400 over hundreds of millions to billions of items [6, 59] and broad applicability across heterogeneous scenarios, our work provides strong theoretical and practical justifications for migrating web-scale vector databases away from dense retrieval and MIPS to Retrieval with Learned Similarities (RAILS) on GPUs.

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

# A  Experiment Setups

## A.1  Reproducibility

Our code will be made publicly available online. Detailed implementations and hyperparameter settings for reproducing our experiment results can be found at the following anonymized GitHub repository: https://anonymous.4open.science/r/rails-4E62, which will be deanonymized after the review process. We discuss specific details below.

## A.2  Parameterization of low-rank ("component-level") embeddings

In this section, we elaborate on the embedding parameterization methods for MoL that we discussed in Section 2.2.

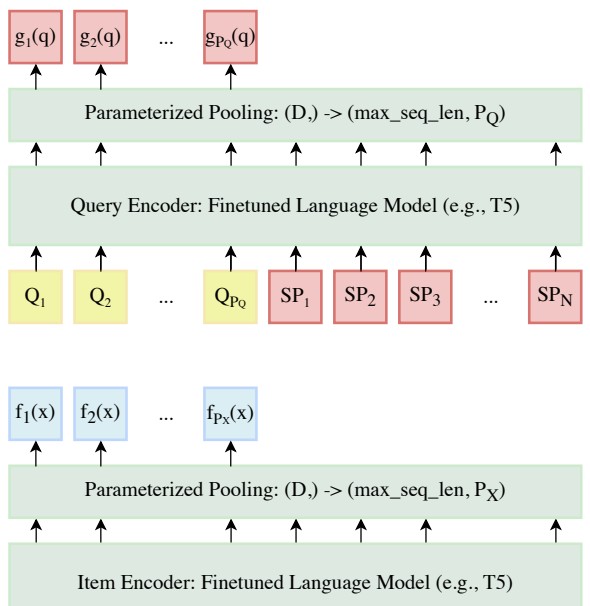
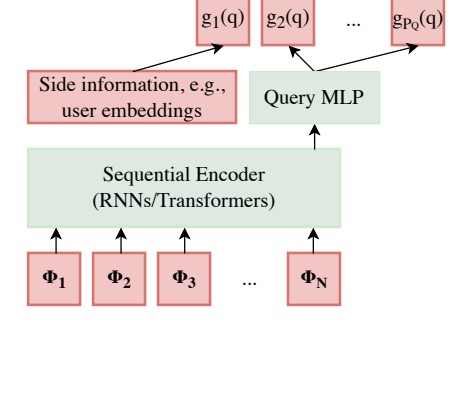
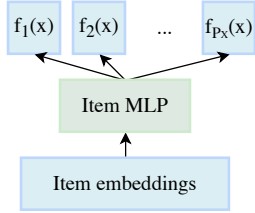

Single, homogeneous feature (Language Models)

Rich, heterogeneous features (Recommendations)

**Figure 3: Illustration of how to parameterize the embeddings to adapt Mixture-of-logits (MoL) learned similarity to various retrieval scenarios, with a language model (LM) finetuning use case in question answering (characterized by a single homogeneous feature) shown on the left, and a recommendation systems use case (characterized by a large number of heterogeneous features) shown on the right. For the Question Answering example on the left, $SP_1, \ldots, SP_N$ represents the original SentencePiece [31] tokens that are inputs to the pre-trained language model LM, e.g., T5 [44]. $Q_1, Q_2, \ldots, Q_{P_Q}$ and $X_1, X_2, \ldots, X_{P_X}$ represent the special aggregation tokens we add to the LM tokenizer for pooling information across the sequence. The "Parameterized Pooling" component uses a $D$-dimensional embedding as input to *parameterize, at an example-level,* how to weight each of the (max_seq_len) encoder outputs for the $P_Q/P_X$ MoL component-level embeddings.**

A.2.1 *Recommendation Systems.* Prior work have shown that careful parameterization of low-rank ("component-level") embeddings, or $f_p(q)$ and $g_p(x)$s for $1 \le p \le P$, can significantly improve MoL's performance [6]. In the context of large-scale recommendation systems, cluster information based on interests of cohorts of members and topics of posts by themselves can lead to 10% recall gain at $K = 400$ [6]. However, we cannot easily access similar information in the publicly available MovieLens [20] and Amazon Reviews [38] datasets. We therefore follow implementation provided by [59] and additionally optionally utilizes a User ID keyed one-hot embedding as one query-side low-rank ("component-level") embeddings $f_p(q)$, which is a widely used technique in recommendation systems [30] that we discussed in Section 2.2. All other component-level embeddings, $f_p(q)$s and $g_p(x)$s, are obtained by applying a multi-layer perceptron (MLP) on top of query-side/item-side representations in standard sequential recommendation setups [22, 27]. The overall setup is illustrated on the right hand side of Figure 3.

A.2.2 *Question Answering (QA).* Unlike Recommendation Systems, retrieval models used in question answering generally take the full semantic representation(s) of the query and/or the document as input, and are finetuned on top of pre-trained language models

with homogeneous inputs, or wordpiece / sentencepiece tokens. Our MoL embedding construction consists of two components, special aggregation tokens and parameterized pooling. We present embedding construction on the query side first.

*Special Aggregation Tokens.* Given both queries and documents are represented as token sequences (e.g., SentencePieces [31] in T5 [44]), we propose to add special tokens that can be used to aggregate different aspects of information as part of the overall self-attention based language model. Specifically, on the query side, let the tokenized sequence be $SP_1, SP_2, \ldots, SP_N$. During finetuning of the pretrained language model, we create $P_Q$ special tokens, $Q_1, \ldots, Q_{P_Q}$, and add them to the vocabulary of the query tokenizer. We also append those exact same $P_Q$ tokens before $SP_1, SP_2, \ldots, SP_N$ [3], so that the $P_Q$ special tokens can be used to aggregate information across the query input using early-fusion mechanisms. Our construction can also be viewed as a way to extend the CLS token in BERT [7, 12] to cover multiple aspects of

---

[3]Note that many question answering scenarios [11, 28, 41, 53, 57] utilize bidirectional language models for retrieval, like BERT [12] or T5 [44]; for recent unidirectional language models, we can add the special aggregation tokens $X_1, \ldots, X_{P_X}$ and $Q_1, \ldots, Q_{P_Q}$ to the end of the input sequence instead.

information, in a way that encourages diversity via the $\mathcal{L}_{MI}$ load balancing loss discussed in Section 2.

*Parameterized Pooling.* We next add a pooling layer after the language model to encourage learning of aggregation mechanisms separate from language semantics. For each position $1 \le p \le P_Q$, this pooling layer defines a probability distribution over different positions in language model's outputs, or $(0, \ldots, max\_seq\_len - 1)$. We further *parameterize* the pooling layer, using the $D$-dimensional embedding at the first position after encoders. This enables us to define a pooling policy, at an example-level, how to weight each of the $max\_seq\_len$ LM encoder outputs to arrive at the $P_Q$ MoL embeddings.

The embedding construction on the item-side is identical. We illustrate the overall finetuning setup we use for question answering on the left hand side of Figure 3.

## A.3   Parameterization of $\pi_p(q, x)$ matrices

We follow the implementation provided in the original MoL paper [59], which parameterizes $\pi_p(q, x)$ as a two-layer multi-layer perceptron (MLP) with SiLU [14] non-linearity. For recommendation datasets (*ML-1M*, *ML-20M*, *Books*), the inputs to this MLP consist of user-side features, item-side features, and the $P$ dot products $\langle f_p(q), g_p(x) \rangle$s between the low-rank embeddings. For question answering datasets (NQ320K), we only use the last part – the $P$ dot products $\langle f_p(q), g_p(x) \rangle$s between the low-rank embeddings – as inputs to this MLP.

## A.4   Hyperparameter settings

*A.4.1   Recommendation Systems.* We use an identical number of sampled negatives for dot product baselines (cosine similarity, "SAS-Rec", "HSTU" rows in Table 3) and Mixture-of-Logits ("SASRec + MoL", "HSTU + MoL" rows in Table 3) to ensure a fair comparison, which is 128 for ML-1M and ML-20M and 512 for Amazon Books following prior work. For "+ MoL" rows, we additionally grid searched $|P_x|$ in $\{2, 4, 8, 16\}$, $d_P$ in $\{32, 64, 128\}$, whether to enable user-id based learned embeddings, and the dropout rate to apply to user-id based embeddings in $\{0.2, 0.5, 0.8\}$ for the smaller MovieLens datasets. We followed initial hyperparameters provided by the authors [59] for all other parameters. The models are trained using PyTorch over 1 NVIDIA RTX 6000 Ada GPU for the smaller *ML-1M* and *ML-20M* datasets and 2 NVIDIA RTX 6000 Ada GPUs for the larger *Books* datasets.

*A.4.2   Question Answering (QA).* We train the model with AdamW optimizer [36], and grid searched learning rate in $\{2e\text{-}4, 5e\text{-}4, 8e\text{-}4\}$ due to the introduction of the parameterized pooling component (Appendix A.2). We apply linear scheduling with warm-up over a fixed 10% of the training epochs. We train the model on 4 NVIDIA H100 80GB GPUs with a local batch size of 512. Note that due to the computational requirements of this dataset, prior work are frequently trained on 8 GPUs [28, 57] or more, e.g., 32 GPUs in GENRE [11] and 256 TPUs in DSI [55]. We perform in-batch negative sampling, consistent with baselines [28, 40]. For MoL hyperparameters, we grid searched $P_Q$ and $P_X$ in $\{(2, 2), (4, 4), (8, 8), (16, 16)\}$, kept $d_P$ identical to the embedding dimension of the pretrained language model (768), and selected the best hyperparameters utilizing a validation set.

