# OpenReview forum: "Efficient Retrieval with Learned Similarities"
_ACM.org/TheWebConf/2025/Conference — WWW 2025 Oral_

### Official Review · Reviewer_6C8f · 2024-11-18

**Novelty:** 7
**Technical Quality:** 5

**Review:**

The authors introduce Mixture-of-Logits (MoL) as a new method to enhance retrieval tasks in recommendation systems and NLP. They demonstrate that MoL can express any learned similarity function. The paper presents both exact and approximate retrieval algorithms and a mutual information-based load-balancing loss designed to improve performance. Experimental results indicate that MoL achieves superb performance across various datasets for recommendation and question-answering tasks.

On the positive side, the paper makes valuable theoretical and practical contributions. The proposed approach is interesting and seems novel, and the authors conducted extensive experiments, considering multiple datasets, various baselines, and different evaluation metrics. They explored the applicability of MoL in two use cases: retrieval in recommender systems and fine-tuning large language models for question answering. Additionally, they studied the effectiveness and efficiency of their approach and conducted ablation tests to evaluate the impact of their proposed loss function. The results show that the proposed methods outperform the considered baselines, including traditional dense, sparse, and generative retrieval methods, with some substantial improvements. The authors also plan to release their code.

On the downside, I found certain sections of the paper, particularly the theoretical proofs, somewhat challenging to follow. Providing additional context and intuition would be beneficial. In addition, for some of the baselines, the authors relied on results published in other papers rather than implementing these baselines themselves. They also did not check whether the performance differences were statistically significant.

**Questions:**

Please see the review above.

**Reviewer Confidence:**

2: The reviewer is willing to defend the evaluation, but it is likely that the reviewer did not understand parts of the paper

**Scope:**

4: The work is relevant to the Web and to the track, and is of broad interest to the community

---

### Official Review · Reviewer_sztF · 2024-11-28

**Novelty:** 6
**Technical Quality:** 6

**Review:**

The paper focuses on improving retrieval through learned similarities. More specifically,
it builds upon the recently proposed Mixture of Logits approach [59], and shows that 1) MoL is a universal similarity approximator  2) how it can be applied to recommender systems and Q&A tasks  3) adding a load balancing regularization loss based on mutual information to improve accuracy.


Strengths:
- the authors prove that MoL is a universal similarity approximator
- they introduce a new regularization loss to improve MoL
- they introduce new approximate for top-K retrieval with MoL
- The paper is overall high-quality, and it focuses on a very important problem, i.e. going beyond dual encoder + dot product retrieval setups while maintaining efficiency. Transition to MoL-based retrieval would have a very big impact in many industrial setups.

Weaknesses:
- the impact of the regularization loss seems small from the ablation study (Tab. 3). Also, there is no reported confidence intervals or statistical testing, which should be added to the experimentation
- the datasets in the experiments are somewhat small
- the latency analysis focuses on MoL setup, it would be nice to include also some baselines
- lack of examples makes the paper a bit harder to follow
- the work is somewhat incremental with respect to [59]. Authors should better clarify the innovation and contributions.

**Questions:**

see weakness above

**Reviewer Confidence:**

3: The reviewer is confident but not certain that the evaluation is correct

**Scope:**

4: The work is relevant to the Web and to the track, and is of broad interest to the community

---

### Official Review · Reviewer_Jyq8 · 2024-12-01

**Novelty:** 6
**Technical Quality:** 6

**Review:**

This paper studies the problem of efficient retrieval with learned similarity function. First, this paper proposes a load balancing loss regularization for mixture of logits, then they show that any high-rank matrix can be decomposed to mixture of logits which means that mixture of logits is a universal approximator. Finally, the study how to use mixture of logits fir retrieving k items from the corpus. They study efficient method for both exact top k search and approximate top k search. Their results show that proposed approach achieves SOTA results in both recommendation and  retrieval tasks. Additionally, they show that their approximate topk approach has a very high recall in comparison to exact topk while it as considerable low latency.

Pros:
1) The paper comes with a strong theoretical proofs to support their findings.
2) the idea is novel and well studied in this paper and make it useful for other methods that utilize learned similarity functions.
3) The experiments show promising results and they performed an extensive ablation study.

Cons:
1) They do not analyse theoritical latency bounds of their approximate top k approach and only do empirical  analysis.

**Questions:**

Can you provide theoretical bounds for latency of you method in comparison with brute force? Or show how much faster it is in theory?

**Reviewer Confidence:**

2: The reviewer is willing to defend the evaluation, but it is likely that the reviewer did not understand parts of the paper

**Scope:**

4: The work is relevant to the Web and to the track, and is of broad interest to the community

---

### Official Review · Reviewer_2Kha · 2024-12-02

**Novelty:** 6
**Technical Quality:** 5

**Review:**

The paper proposes an approach based on a mixture of logits to learn the similarities between queries and documents/items.

The description of the method is not sufficiently detailed. For example, the concept of "P groups of low-rank embeddings" is not defined. Similarly, "adaptive gating weights" should be defined better: if it is basically the dot product (as suggested in the line after) this should be stated more clearly. What does the s stands for in <fp(q), gp(x)>s?

While the results are numerically promising, there is no statistical testing.

As a minor, the authors should harmonize the number of significant digits in their tables: it is strange to see tables with different numbers of significant digits.

**Questions:**

The description of the method is not sufficiently detailed. For example, the concept of "P groups of low-rank embeddings" is not defined. Similarly, "adaptive gating weights" should be defined better: if it is basically the dot product (as suggested in the line after) this should be stated more clearly. What does the s stands for in <fp(q), gp(x)>s?

**Reviewer Confidence:**

3: The reviewer is confident but not certain that the evaluation is correct

**Scope:**

3: The work is somewhat relevant to the Web and to the track, and is of narrow interest to a sub-community